# Effect of Rosehip Powder Addition on Dough Extensographic, Amylographic and Rheofermentographic Properties and Sensory Attributes of Bread

Nicoleta Vartolomei and Maria Turtoi *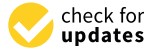

Cross-Border Faculty, Dunarea de Jos University of Galati, 47 Domneasca Street, 800008 Galati, Romania; nicoleta.vartolomei@ugal.ro
* Correspondence: maria.turtoi@ugal.ro

**Abstract:** One of the improvers used in breadmaking is ascorbic acid (AA), a chemical compound that strengthens the dough and extends the shelf life of bread. This work investigates the suitability of replacing the synthetic AA with rosehip powder (Rp) rich in this bioactive compound. Thus, a comprehensive study of wheat flour (WF) replaced with 0.5–2.5% *w/w* Rp regarding the extensographic, amylographic and rheofermentographic properties of dough and sensory attributes of bread was performed. WF without RP or AA addition of 2 mg/100 g was used as a control. A sample with an AA addition of 2 mg/100 g was also used. The Rp addition positively influenced the extensographic, amylographic and rheofermentographic properties of the dough. The dough resistance to extension, R, in Brabender Units (BU), increased from 330 ± 1.41 BU (control) to 995 ± 1.41 BU (2.5% *w/w* Rp) for a resting time of 90 min. The gelatinization temperature of the dough increased from 61.0 °C (control) to 62.9 °C (2.5% *w/w* Rp). The volume of gases retained in the dough increased in bread with up to 2.0% *w/w* Rp and afterwards decreased. The sensory properties of the bread, e.g., external appearance, volume, flavor, and taste, were appreciated by the sensory panel and received higher total scores than the control bread. According to the results presented in this work, the optimum concentration of Rp was 1.5% *w/w*. However, because the concentration of AA in Rp at the moment of use could vary, it would be better to consider an optimum range, e.g., 1.0–2.0% *w/w* Rp. The study showed that the Rp is appropriate for breadmaking as an alternative to synthetic AA.

**Keywords:** amylograph; ascorbic acid; bread; dough; extensograph; rheofermentometer; rosehip powder; sensory attributes; wheat flour



## 1. Introduction

Ascorbic acid (AA) is an antioxidant compound with various uses, such as in medicine [1], dietary supplements [2] (p. 962), and as a food additive in the food industry [3] (p. 284). It is obtained as L-AA by chemical synthesis, starting from glucose [4], and approved as a food additive/preservative in the EU with the number E300 [5,6], USA [7], Canada [8], Australia, and New Zealand [9].

AA is also naturally present in plants, especially in fruits and vegetables, as the reduced form of vitamin C [10]. The amount of vitamin C in plants depends on many factors, such as the diversity of plants, conditions of soil and climate, duration and storage conditions, and preparation methods [10]. Among the well-known sources of vitamin C are citrus fruits, kiwifruit, tomatoes, broccoli, peppers, strawberries, Brussels sprouts, and cantaloupe [11]. According to USDA National Nutrient Database [12], vitamin C is abundant in several raw plant sources, e.g., Kakadu plum (1000–5300 mg/100 g), camu camu (2800 mg/100g), acerola (1677 mg/100 g), sea buckthorn (695 mg/ 100 g), Indian gooseberry (445 mg/100 g), rosehip (426 mg/100 g), guava (228 mg/100 g), blackcurrant (200 mg/100g), yellow bell pepper (183 mg/100 g), red bell pepper (128 mg/100 g), kale (120 mg/100 g), etc. Vitamin C is soluble in water and sensitive to heat. Therefore, extended

storage and cooking of fruits and vegetables rich in this compound may cause a reduction in vitamin C content [12].

AA is used in breadmaking as an oxidizing agent to obtain bread and other bakery products with desired and consistent quality [13]. It is added in small amounts into flour or the dough and contributes to dough stabilization, preserving its elasticity and consistency and allowing better gas retention [14,15]. Additionally, AA permits faster proofing and bread with a softer crumb and longer freshness [16].

The output of AA increased in the last decades and China became the world leader, providing around 80% of the quantity required for the food and beverages industry, pharmaceutical, beauty and personal care, and animal feed [17]. Synthetic AA received the status generally recognized as safe (GRAS) [7]. Additionally, the L-AA production and utilization comply with good manufacturing practices (GMP) [7,18].

Nowadays, there is an increase in demand for AA, e.g., for the production of functional food and beverages [18]. Additionally, the concerns of consumers regarding the use of food additives and the consumption of food with high levels of synthetic additives determined the current trend to reduce the use or abandon artificial food additives and the consumer demand for natural [19], fresh, minimally processed foods [20], and foods with fewer additives and preservatives [21]. These trends led to the need to find other sources of AA, such as natural sources highly available [9]. Thus, research focused on replacing synthetic AA with AA-rich plant materials was initiated, disseminated, and applied in practice [22–24].

The first known research was performed at Campden BRI, UK, using an extract of acerola cherry fruits (*Malpighia emarginata*) in breadmaking to replace synthetic AA in white wheat flour. The dough obtained had the resistance to deformation of 505 Brabender Units (BU), similar to that of the dough with AA (495 BU), and the extensibility of 147 mm, compared to 169 mm for the dough with AA and 179 mm for the control. Additionally, the properties of bread, i.e., the volume of loaf bread, the structure of the crumb, and the texture, were comparable to control. The results seem promising even if only part of the experimental data were disclosed [22] (pp. 327–329).

Some recent research highlighted the use of acerola powder as a substitute for AA in breadmaking. Bourekoua et al. [23] added acerola powder (0–5% *w/w*) to rice flour to study the influence on the properties of gluten-free bread. Thus, the volume, size and area fraction of cells and the antioxidant activity increased in bread with acerola powder. Additionally, the textural properties were improved through the decrease in firmness and chewiness and an increase in springiness. The sensory attributes allowed the authors to conclude that an optimum level of acerola powder addition would be 3% *w/w*. The authors of [24] assessed the acerola powder as a replacement for AA and an improver in bread produced using white and whole meal WF. Acerola powder behaved similarly to AA: it enhanced the dough development time, softening degree, and stability, increased the specific volume of white wheat bread, and reduced the hardness of all pieces of bread, keeping the crust and crumb color unchanged.

The plant materials rich in AA [12] include the fruits of the wild rose shrub. The rosehip plant (*Rosa canina* L.) is native to Europe, Northwest Africa, and Western Asia [25]. The climbing shrub is also found in North America, where it was introduced in colonial times, Australia and New Zealand [26]. Its pomaceous pseudo fruits, called rosehip or rose haw, are a valuable source of vitamin C [27], e.g., 100–1.400 mg/100 g and medium values of 0.4–0.8% in fresh fruits [28,29]. Humans have always consumed rosehip fruits as an essential part of a healthy diet, long before the word vitamin was invented [30] and used these fruits as herbal medicine without knowing how human health is affected [31,32]. Additionally, the bakers have used rosehip fruits in powder or extract form to increase the nutritional value of bread [33–35]. Another application is an oil extracted from rosehip fruits to improve *Aloe vera* gels used as postharvest coatings of stone fruits [36,37].

This work continues the original research presented by Vartolomei & Turtoi [38], which studied the use of rosehip powder to replace the synthetic AA in wheat flour (WF) and

observed its influence on the farinographic properties of the dough and the quality of bread. Thus, the research aimed to investigate the impact of 0.5–2.5% *w/w* Rp addition on the extensographic, amylographic and rheofermentographic properties of dough and the sensory attributes of bread.

## 2. Materials and Methods

### 2.1. Materials

The white wheat flour (WF, type 550), non-treated with improvers, was provided by a local milling and baking company (Dizing SRL, Brusturi, Neamt County, Romania). The proximate composition of WF was: moisture 14.15 ± 0.06%, ash 0.55 ± 0.01%, proteins 13.45 ± 0.03%, wet gluten 34.10 ± 0.16%, carbohydrates 70.68 ± 0.29%, and fibers 0.10 ± 0.07% [38].

Rosehip powder (Rp) was obtained as described in [38] and had the following proximate composition: moisture 13.40 ± 0.15%, ash 6.50 ± 0.07%, proteins 4.89 ± 0.11%, carbohydrates 73.66 ± 0.19%, fibers 8.63 ± 0.12%, and vitamin C 420 ± 16.09 mg/100 g [26].

Compressed yeast (30% d.w., Pakmaya, Rompak SRL, Pașcani, Iasi County, Romania) and salt (Slanic Saltwork) were procured from the local market.

Enzymes and Derivates, Costișa, Neamt County, provided the synthetic AA (powder, min. purity 99%).

### 2.2. WF-Rp Mixtures Preparation

Mixtures of WF with 0.5%, 1.0%, 1.5%, 2.0%, and 2.5% *w/w* Rp preparation, analysis following the official methods [39–41], and proximate composition were presented and discussed in [38]. The content of vitamin C in flour mixtures, in mg/100 g WF-Rp, was 2.0 ± 0.20 (0.5% *w/w* Rp), 4.0 ± 0.36 (1.0% *w/w* Rp), 6.0 ± 0.30 (1.5% *w/w* Rp), 8.0 ± 0.17 (2.0% *w/w* Rp), and 10 ± 0.26 (2.5% *w/w* Rp) [38].

### 2.3. Extensographic Measurements

The dough extensographic properties were determined using a Brabender Extensograph Model E3 (Duisburg, Germany), following the AACC method no. 54-10.01 [40] and SR EN ISO 5530-2:2015 [41]. The samples were prepared from flour, distilled water, and salt in the Farinograph [41] to ensure objectivity and reproducibility during dough preparation and a constant starting consistency. The dough was then stretched until rupture in the Extensograph-E3 and the applied force was measured and recorded online on the monitor as a color diagram named extensogram. The procedure was repeated three times for 30, 60 and 90 min, and in duplicate.

The extensogram represents the applied force as a function of the stretching time (in min) and stretching length (in cm) and provides data for the resistance to extension, extensibility, maximum resistance to extension, ratio number, maximum ratio number, and dough energy (area below the curve) [42].

### 2.4. Amylographic Measurements

The Brabender Amylograph Model E (Duisburg, Germany) was used to determine the $\alpha$-amylases activity in WF and flour mixtures according to the AACC method no. 22-10 [37] and SR EN ISO 7973:2016 [41]. A suspension of flour and water was heated in the rotating container of the Amylograph-E at a rate of 1.5 °C per minute. Heating simulates what is happening inside the crumb during the baking process. A probe introduced into the sample was diverted into the Amylograph chamber as a function of the viscosity of the sample. The deflection of the probe was measured as a torque value and instantly recorded in the amylogram [43].

The parameters visualized in the torque curve of the amylogram are the initial temperature (when gelatinization starts), the maximum gelatinization temperature, and the maximum viscosity of gelatinization [43].

## 2.5. Rheofermentographic Measurements

Dough characteristics during proofing were measured with a Chopin Rheofermentometer Model Rheo F3 (Chopin Technologies, Villeneuve-la-Garenne, France) following the AACC method no. 89-01 [40]. The measurements included the following steps: yeast pretreatment, dough mixing, and gas production determination.

Dough formula. The formula of the dough was 100 g WF of wheat mixtures, 2 g salt, 4.06 g compressed yeast, and warm distilled water. The water volume was correlated to the water absorption of flour: 58–62 mL/100 g flour [38]. The formula based on 100 g of flour can be upgraded to obtain a suitable quantity of dough.

Pretreatment of yeast. The compressed yeast was weighed accurately, crumbled into a beaker, and rehydrated in 25 mL of distilled water at $21 \pm 1\,^\circ$C five (5) min before mixing.

Mixing of dough. The dough was prepared in the bowl of a dough mixer (Tower T12039 Rose Gold, 2.5 L, 300 W). All dry ingredients were weighed, added to the bowl, and mixed with the yeast suspension. The remaining water was used to rinse the beaker and poured into the bowl. The water temperature was adjusted to obtain the dough at $30 \pm 1\,^\circ$C. The dough was mixed for 8–10 min until full development, then rested for 5 min while the temperature was measured.

Determination of gas production. The dough was introduced in an airtight container and subjected to fermentation in a temperature-controlled environment ($30 \pm 1\,^\circ$C). The test duration was 180 min. Gases were produced during fermentation due to yeast activity. The pressure inside the container was measured every 45 s. Additionally, a sensor above the dough showed its development and stability to determine the optimum baking time. Thus, a single automated test lasting for three hours (180 min) determined dough development, gas production due to yeast action, dough porosity, and dough tolerance during proofing. The results were delivered under a diagram called rheofermentogram, which contains two curves and all the calculations. The upper one, the dough development curve, reveals the maximum dough height ($H_m$), time to reach the maximum height ($T_1$), final dough height ($h$), and time of relative stabilization. The lower one, the gas release curve, was obtained through a direct cycle, in which the device measured the total gas production, and an indirect cycle, which measured the gas retention, i.e., the porosity of the dough [44].

## 2.6. Breadmaking Procedure

The bread was obtained using the facilities of the Dizing breadmaking company and the recipe and procedure described in Vartolomei & Turtoi [38]. Bread samples were analyzed after storage for two hours at room temperature. The physicochemical parameters—bread height, bread volume, moisture, porosity, elasticity, and acidity of crumbs—were presented in [38] and the sensory attributes are discussed in this work.

## 2.7. Sensory Analysis

A panel of 21 assessors (7 men and 14 women aged between 20 and 50 years) were selected from staff members of the Dizing breadmaking company and trained in advance. The selection criteria included, in addition to age, non-smoking, no known allergies, and no difficulties perceiving flavor, taste or swallowing foods. Each participant signed an informed consent letter. The Ethics Committee of Dizing company approved the study.

The standard SR 91:2007 was followed for the sensory analysis [41]. The 20–point consumer test was applied. The method involved evaluating the sensory attributes of the bread, giving them points such that the highest score was 20. Thus, there were 3 points for external appearance/shape and volume, crust appearance, consistency and chewing behavior of the crumb, and flavor, and 4 points for crumb appearance and taste.

The test was performed during the morning, in the time interval 10:00–11:00, and in an isolated room with good lighting and natural ventilation. After cooling, the bread loaves were cut into 20 mm thick slices (approximately 60 g, including the crust and crumb) and presented to assessors in glass vessels with lids. The samples were 2-digit coded and served in a randomized and balanced order. The assessors evaluated each sample by

gently chewing it in the oral cavity, having enough time to perceive the flavor of the bread thoroughly before and after swallowing and rinsing the palate with drinking water after each evaluation.

### 2.8. Statistical Analysis

All experiments were performed in duplicate and the average values with standard deviations are presented. Microsoft Excel program, Microsoft Office 2010, was used to carry out a one-way analysis of variance (ANOVA) to detect significant differences among results. Tukey's honestly significant difference (HSD) method was applied at a 99.0% confidence level using the Statgraphics Centurion XVI.I software (Statgraphics Technologies, Inc., The Plains, VA, USA).

## 3. Results and Discussion

### 3.1. Extensographic Properties

The extensographic properties supplied by extensograms are presented and discussed as follows.

#### 3.1.1. Resistance to Extension

The resistance to extension, $R_{50}$, in BU, is obtained after 5 cm (50 cm) of stretching length, while the maximum resistance to extension, $R_{max}$, in BU, is measured at the peak of the curve. Figure 1 depicts the $R_{50}$ (Figure 1a) and $R_{max}$ (Figure 1b) for 30, 60, and 90 min and all samples (control, samples with 0.5–2.5% $w/w$ Rp, and sample with 2 mg AA/100 g).

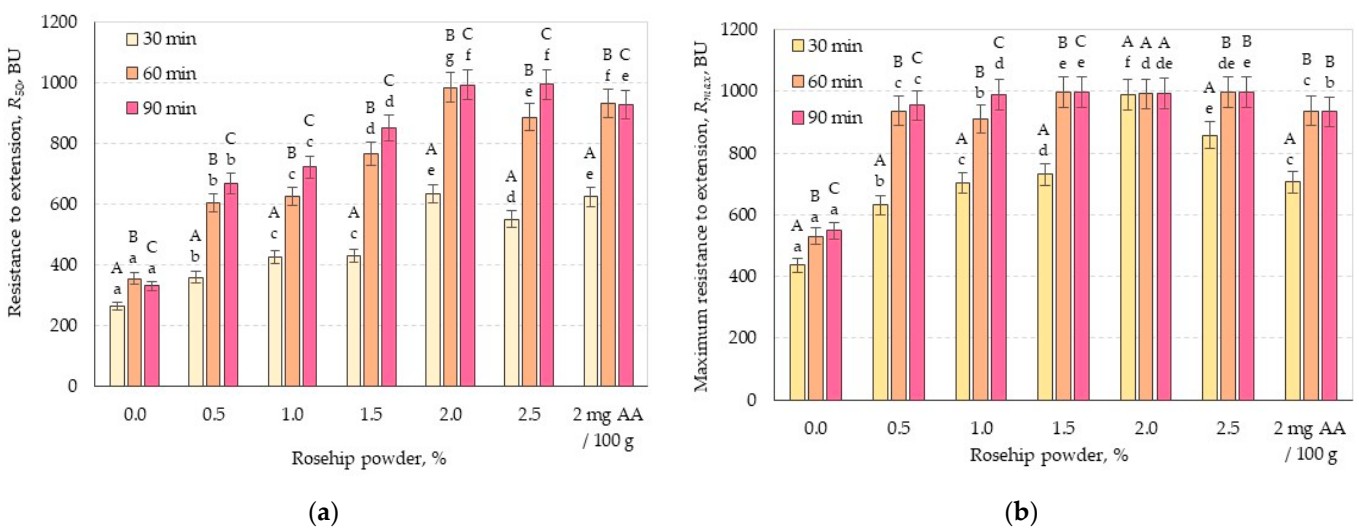

(**a**)                                                                 (**b**)

**Figure 1.** Resistance to extension as a function of Rp addition: (**a**) Resistance to extension at 50 mm ($R_{50}$); (**b**) Maximum resistance to extension ($R_{max}$). Different capital letters represent statistically different mean values ($p \leq 0.01$) of each sample for different times (30, 60, and 90 min). Different lower-case letters are responses statistically different ($p \leq 0.01$) of all samples for each time (30, 60, and 90 min). BU—Brabender Units, AA—ascorbic acid.

The Rp addition determines the increase of the resistance to extension, $R_{50}$, of each sample for all durations (30, 60, and 90 min); only the $R_{50}$ of the control at 90 min (330 ± 1.41 BU) makes an exception since it is lower than the value at 60 min (355 ± 1.41 BU). According to the Tukey HSD test, mean values of $R_{50}$ are statistically different ($p \leq 0.01$) except for the sample with 2.0% $w/w$ Rp for all durations and the sample with 2.5% $w/w$ Rp for 60 and 90 min. The values of $R_{50}$ for the mixture with 2.0% $w/w$ Rp at 60 min (985 ± 0.71 BU) and 90 min (993 ± 0.71 BU), and for the mixture with 2.5% $w/w$ Rp at 90 min (995 ± 1.41 BU) are close to 1000 BU, the highest value recorded by Brabender E3. The $R_{50}$ values of all samples

for the same duration increase until 2.0% $w/w$ Rp, then slightly decrease. According to the Tukey HSD test, most of the mean values are statistically different ($p \leq 0.01$).

The $R_{max}$ (Figure 1b) has a similar evolution to the $R_{50}$ but with greater values. The values obtained for the control are $435 \pm 1.41$ UB at 30 min, $530 \pm 1.41$ UB at 60 min and $547 \pm 1.41$ UB at 90 min. For some samples, $R_{max}$ is located on the flattened peak of the extensographic curves. It reaches or is near the maximum of 1000 UB at which the device is set. Values higher than 1000 UB are impractical since a dough with such $R_{max}$ is unsuitable for breadmaking. Thus, $R_{max}$ values are $988.5 \pm 0.71$ BU, $991.5 \pm 0.71$ BU, and $993.5 \pm 0.71$ BU at 30, 60 and 90 min for 2.0% $w/w$ Rp, which are slightly lower than $999.0 \pm 0.00$ BU and $998.0 \pm 0.00$ BU at 60 and 90 min for 1.5% $w/w$ Rp and $995.0 \pm 1,41$ BU at 60 and 90 min for 2.5% $w/w$ Rp. It appears as though the maximum is situated between 1.5% and 2.5% $w/w$ Rp. According to the Tukey HSD test, most of the mean values are statistically different ($p \leq 0.01$).

Gómez et al. [45] obtained similar results by adding 25 and 5% fibers from peas, cocoa, coffee, oranges, wheat, and microcrystalline cellulose. They showed that fiber additions enhanced the dough tolerance and stability independently of the fiber source.

### 3.1.2. Extensibility

The extensibility of the dough, $E$, in mm, is shown in Figure 2. According to the Tukey HSD test, most mean values are statistically different ($p \leq 0.01$), except for the control. The $E$ values are $205 \pm 1.41$ mm, $191 \pm 1.41$ mm, and $204 \pm 2.83$ mm at 30, 60, and 90 min for the control. The $E$ slightly decreases at 30 min after Rp exceeds 1.5% $w/w$. The decrease is more pronounced at 60 and 90 min for the Rp values higher than 0.5% $w/w$ compared to 30 min. The dough obtained from WF with 2 mg AA/100 g showed the most considerable reduction of E values compared to the control.

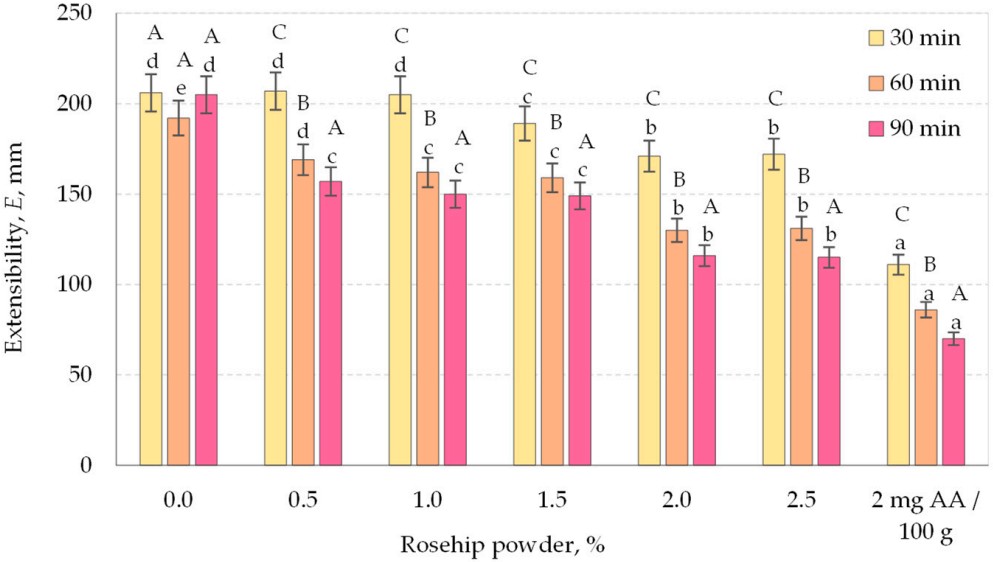

**Figure 2.** Extensibility ($E$, mm) as a function of Rp addition. Different capital letters represent statistically different mean values ($p \leq 0.01$) of each sample for different times (30, 60, and 90 min). Different lower-case letters are responses statistically different ($p \leq 0.01$) of all samples for each time (30, 60, and 90 min). AA is ascorbic acid.

The results are similar to the reports of another research. Thus, Koletta et al. [46] reported a reduction in the extensibility of the dough obtained from refined WF substituted with 60% of whole rye flour, whole barley flour, and oat flour, versus control. Other studies, however, have shown different results. For example, the authors of [47] added 10%, 20%, and 30% chickpea flour to WF and observed a decrease of $R_{50}$ and $E$. They also reported a slight increase of $R_{50}$ and $E$ with the resting time (45, 90 and 135 min). If for

10% chickpea flour, the $R_{50}$ and $E$ values did not decrease much, significant differences were reported in the dough (sticky and soft) and bread (lower volume) characteristics for higher additions. The authors found a weaker gluten network for more than 20% chickpea flour and concluded that the decrease in the gluten content caused the reduction of $R_{50}$ and $E$ and the worsening of bread characteristics.

### 3.1.3. Ratio Numbers

The ratio number (quality index) $R_{50}/E$ and maximum ratio number Rmax/E are presented in Figure 3. Their values increase proportionally with the addition of Rp and resting time. The increases are higher for 60 min resting time versus 30 min and smaller for 90 min compared to 60 min. The sample with AA showed the highest $R_{50}/E$ and $R_{max}/E$ values, mainly due to the lowest extensibility. According to the Tukey HSD test, most mean values are statistically different ($p \leq 0.01$).

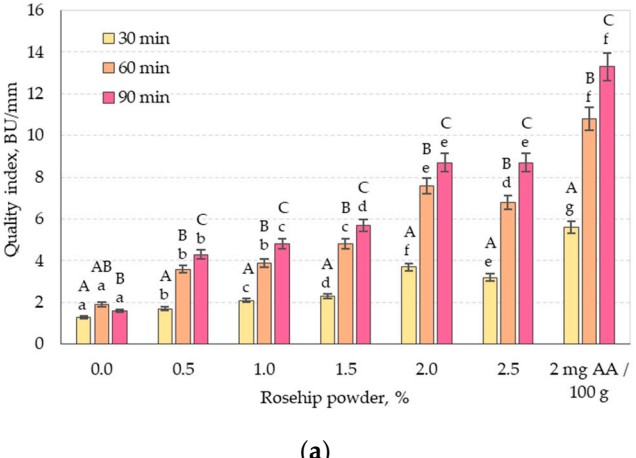

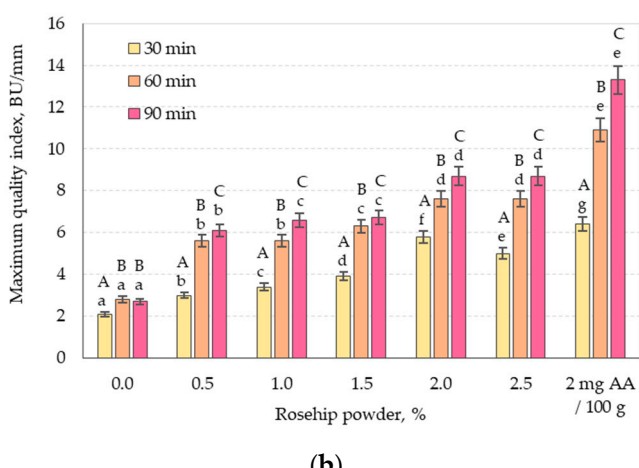

(**a**)          (**b**)

**Figure 3.** Ratio numbers as a function of Rp addition: (**a**) Ratio number, $R_{50}/E$, in BU/mm; (**b**) Maximum ratio number, $R_{max}/E$, in BU/mm. Different capital letters represent statistically different mean values ($p \leq 0.01$) of each sample for different times (30, 60, and 90 min). Different lower-case letters are responses statistically different ($p \leq 0.01$) of all samples for each time (30, 60, and 90 min). BU—Brabender Units, AA—ascorbic acid.

Similar results are reported in the literature. The authors of [46] communicated an increase in the $R_{50}/E$ index due to the decrease of $E$ when replacing 60% of WF with whole rye, whole barley, and oat flours. Other authors [45] revealed higher values of the $R_{50}/E$ index when adding 2 and 5% fibers from peas, cocoa, coffee, oranges, wheat, or microcrystalline cellulose.

In contrast, the authors of [47] observed that the $R_{50}/E$ index decreased with the rise of chickpea flour addition and increased with the resting time of 45, 90 and 135 min in doughs obtained from WF with 10%, 20%, and 30% chickpea flour. They suggested that the oxygen supplied to the dough by kneading oxidized the –SH radicals and the SH bonds were transformed into disulfide bonds (–S–S–). Then, these newly formed bonds were involved in the strengthening of gluten and dough by increasing the elasticity, thus decreasing the $E$ and increasing the $R_{50}$ of the dough.

### 3.1.4. Dough Energy

The energy of the dough $E_d$, in cm$^2$, depending on the addition of Rp, shown in Figure 4, represents the force necessary to produce the dough deformation during stretching. The Brabender E3 device software calculates the energy as the area below the extensographic curve, multiplying the stretching force of the dough in BU by the stretching length. Its value varies in direct proportion to the resilience of the dough; therefore, the dough has a greater tolerance if the energy value is higher.

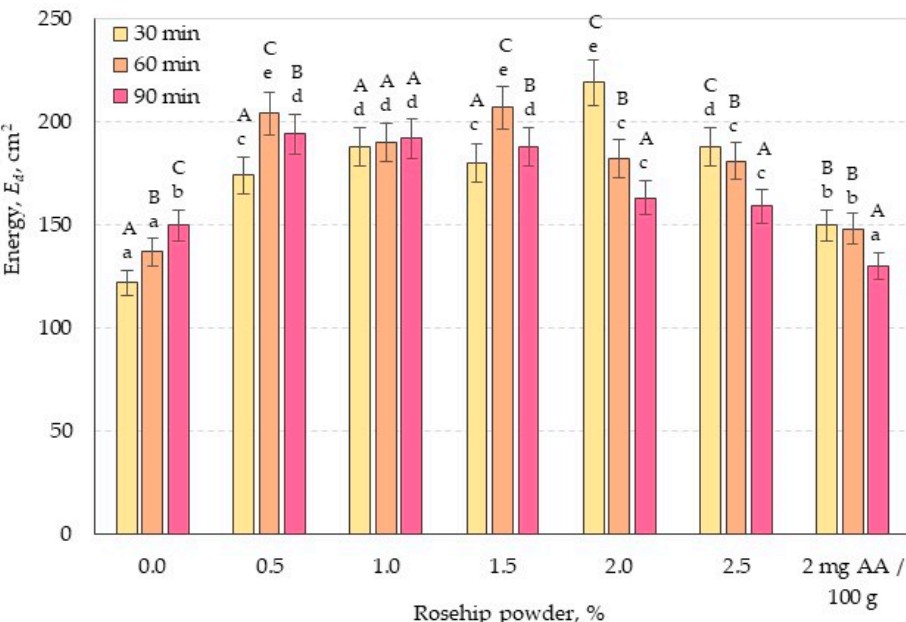

**Figure 4.** Dough energy ($E_d$, cm$^2$) as a function of Rp addition. Different capital letters represent statistically different mean values ($p \leq 0.01$) of each sample for different times (30, 60, and 90 min). Different lower-case letters are responses statistically different ($p \leq 0.01$) of all samples for each time (30, 60, and 90 min). AA is ascorbic acid.

According to the Tukey HSD test, most mean values are statistically different ($p \leq 0.01$). An increase in dough energy with the Rp addition of 0.5–2.0% $w/w$ is followed by a decrease when 2.5% $w/w$ Rp is added. Additionally, the dough with 2 mg AA/100 g needs less energy to be stretched than any dough with an Rp addition. The smallest increase of dough energy in this sample with AA addition is due to the lowest value of $E$.

A thorough analysis of all extensographic parameters, resistance to extension, extensibility and energy shows that doughs obtained from a flour mixture with less than 2.0% $w/w$ Rp addition have the best behavior during the extensographic tests. The increase in energy for samples with added Rp is, on average, about 50% compared to the control. Correlating this with the content of vitamin C in the Rp ($420 \pm 16.09$ mg/100 g) and the flour mixtures (2.0–10.0 mg/100 g), additions above 2.0% $w/w$ Rp could be more than necessary for better behavior of the dough.

The results differ from those of other research mainly because the replacements of WF and their values varied. Thus, the authors of [46] reported lower energy of the dough necessary for stretching it to break down when adding 60% whole rye flour, whole barley flour, and oat flour to WF and correlated the energy decrease with reduced gluten-forming proteins due to additions. Additionally, the dough sample obtained from wheat with oat flour needed the least energy because gluten weakened the network formed by gluten the most.

In another study [47], the authors reported a decrease in the energy of dough for several additions of chickpea flour to WF (10%, 20%, and 30%) and resting times (45, 90, and 135 min). They emphasized that the presence of specific enzymes able to interact with gluten heavily contributed to worsening the rheological characteristics, including the energy of the dough.

Similarly, Nikolić et al. [48] observed that dough energy decreased from $67.8 \pm 3$ cm$^2$ to $32.4 \pm 2$ cm$^2$ when buckwheat flour was added to WF for all additions (3%, 5%, 10%, 15%, and 20%), simultaneously with the decrease in $R_{50}$ from $345 \pm 15$ BU to $222 \pm 10$ BU, $E$ from $126 \pm 10$ BU to $106 \pm 5$ BU and ratio number from $2.5 \pm 0.2$ to $2.1 \pm 0.2$ BU/mm. The buckwheat flour reduced gluten-forming proteins in the flour mixtures. No enhancers were used in the study.

The authors of [49] added 250 and 500 ppm (25 mg and 50 mg/100 g) L-AA to WF mixtures with 10% and 50% soybean flour. The control WF with L-AA addition experimented with an $R_{max}$ that exceeded 1000 BU (the curve flattened at the maximum value recorded by the extensograph, which means that the doses of L-AA used were too high for WF). In contrast, the $R_{max}$ was improved for all soybean flour mixtures with L-AA additions, compared to controls with or without L-AA. The dough $E$ was also enhanced, except for WF mixtures with 50% soybean meal and 250 ppm AA addition, which decreased after 135 min of rest.

In conclusion, the use of 0.5–2.5% $w/w$ Rp in WF has a positive influence on the extensographic parameters, increasing the resistance to extension, ratio number and dough energy but showing that Rp additions above 1.5% $w/w$ lead to $R_{50}$ and $R_{max}$ higher than the maximum recorded by the extensograph which indicates a dough unsuitable for breadmaking.

### 3.2. Amylographic Properties

The Brabender Amylograph device provided the results of the amylographic test in a diagram called amylogram containing a curve which represents the variation of the viscosity of the flour gel formed versus time and temperature in amylographic units (AU). The values of the key rheological properties of the gel are shown in Table 1.

**Table 1.** Amylographic parameters of dough obtained from flour mixtures WF-Rp.

| Parameters / Samples | Temperature at the Beginning of Gelatinization, $T$ °C | Maximum Temperature of Gelatinization, $T_{max}$ °C | Maximum Gelatinization Viscosity, AU |
|---|---|---|---|
| WF (Control) | $61.00 \pm 0,00$ [a] | $84.25 \pm 0.00$ [a] | $556.0 \pm 2.0$ [b] |
| WF–Rp 0.5% $w/w$ | $61.50 \pm 0.10$ [a,b] | $85.70 \pm 0.10$ [b] | $617.5 \pm 1.5$ [d] |
| WF–Rp 1.0% $w/w$ | $61.55 \pm 0.05$ [a,b] | $85.85 \pm 0.00$ [b,c] | $584.0 \pm 1.0$ [c] |
| WF–Rp 1.5% $w/w$ | $61.60 \pm 0.10$ [b] | $86.05 \pm 0.00$ [b,c,d] | $566.0 \pm 2.0$ [b] |
| WF–Rp 2.0% $w/w$ | $62.35 \pm 0.05$ [c] | $86.25 \pm 0.00$ [c,d] | $537.5 \pm 1.5$ [a] |
| WF–Rp 2.5% $w/w$ | $62.90 \pm 0.10$ [c] | $86.40 \pm 0.00$ [d] | $534.0 \pm 1.0$ [a] |
| WF–AA 2 mg/100 g | $61.55 \pm 0.05$ [a,b] | $86.10 \pm 0.10$ [b,c,d] | $573.5 \pm 2.5$ [e] |

WF—wheat flour, Rp—rosehip powder, WF–Rp—mixtures of wheat flour with rosehip powder, WF–AA—wheat flour with ascorbic acid addition of 2 mg/100 g. Different letters on columns represent statistically different mean values ($p \leq 0.01$).

Replacing the WF with Rp modified the gel properties. Thus, the gelatinization temperature increased from $61.0 \pm 0.0$ °C (control) to $62.90 \pm 0.10$ °C for 2.5% $w/w$ Rp. Its value for the WF-AA sample was $61.55 \pm 0.05$ °C, identical to that of the sample with 1.0% $w/w$ Rp. The temperature at the peak of the amylographic curve has the same behavior, increasing from $84.25 \pm 0.00$ °C for the control to $86.40 \pm 0.00$ °C for 2.5% $w/w$ Rp addition. A different evolution was obtained for the viscosity at the peak of the amylogram. It increased from $556.0 \pm 2.0$ AU for the control to $617.5 \pm 1.5$ AU for 0.5% $w/w$ Rp, then decreased for the following Rp values, reaching lower values than for the control when 2.0% and 2.5% $w/w$ Rp were used (Table 1). The maximum gelatinization viscosity in the sample with 2 mg AA/100 g was $973.5 \pm 2.5$ AU. According to the Tukey HSD test, most mean values are statistically different ($p \leq 0.01$).

Other additions to WF led to comparable results. Thus, Nikolić et al. [48] and Yoo et al. [50] obtained higher values for the maximum gelatinization temperature and the viscosity at maximum gelatinization when adding buckwheat flour to WF. The maximum viscosity value was almost double at the highest addition of 20% buckwheat flour [48]. In another research [51], when WF was substituted with 10%, 20%, 30%, 40%, and 50% whole buckwheat flour for chapati, the gelatinization temperature raised by almost nine Celsius degrees and the peak viscosity decreased as a result of reduced starch content and increased dietary fibers content. Additionally, the amylographic parameters of the amylogram peak— maximum temperature and gel viscosity—increased for the dough obtained from mixtures

of WF and 5% to 30% *Boletus edulis* mushrooms flour due to increased protein and lipid content derived from mushrooms flour [52].

In conclusion, substituting the WF with 0.5–2.5% *w/w* Rp modified the gel properties and indicated good enzyme activity in the dough.

### 3.3. Rheofermetographic Properties

The Chopin Rheo F3 device measures the properties of the dough during proofing and provides diagrams called rheofermentograms for each rheofermentographic test. A rheofermentogram comprises two curves: the upper one, the dough development curve and the lower one, the gas production curve. The rheofermentograms provide specific results for each curve which are presented and discussed in the following subchapters.

#### 3.3.1. Properties from the Dough Development Curve

The properties of the dough resulting from the analysis of the dough development curve are the dough height at the peak of the curve ($H_m$) the time $T_1$ to reach $H_m$, the final height of the dough ($h$), and the time of relative stabilization ($T_2 - T_2'$) at the maximum point. Additionally, the rheofermentogram indicates the calculated value of the percentage of fall (Table 2).

**Table 2.** Rheofermetographic properties of dough resulted from the dough development curve.

| Parameters / Samples | Maximum Development (Dough Height), $H_m$, mm | Time Required to Reach $H_m$, $T_1$, min | Time of Relative Stabilization | | Final Height of the Dough, $h$, mm | Percentage of Fall $(H_m - h)/H_m$, % |
|---|---|---|---|---|---|---|
| | | | $T_2$, min | $T_2'$, min | | |
| WF (Control) | 64.6 ± 0.14 [b] | 96 ± 1.41 [a] | 147 ± 1.41 | 67 ± 1.41 | 60.9 ± 0.14 [a] | 5.73 ± 0.01 [c] |
| WF–Rp 0.5% *w/w* | 61.4 ± 0.14 [a] | 178 ± 1.41 [b] | – | – | 61.3 ± 0.14 [a] | 0.16 ± 0.00 [a] |
| WF–Rp 1.0% *w/w* | 65.3 ± 0.14 [b] | 178 ± 1.41 [b] | – | – | 65.2 ± 0.14 [b] | 0.15 ± 0.00 [a] |
| WF–Rp 1.5% *w/w* | 70.0 ± 0.14 [e] | 180 ± 0.00 [b] | – | – | 70.0 ± 0.14 [d] | 0 [a] |
| WF–Rp 2.0% *w/w* | 71.3 ± 0.14 [f] | 180 ± 0.00 [b] | – | – | 71.3 ± 0.14 [e] | 0 [a] |
| WF–Rp 2.5% *w/w* | 69.1 ± 0.14 [d] | 180 ± 0.00 [b] | – | – | 69.1 ± 0.14 [d] | 0 [a] |
| WF–AA 2 mg/100 g | 67.6 ± 0.14 [c] | 176 ± 1.41 [b] | – | – | 66.5 ± 0.28 [c] | 1.63 ± 0.21 [b] |

WF—wheat flour, Rp—rosehip powder, WF–Rp—mixtures of WF with Rp, WF–AA—WF with ascorbic acid addition of 2 mg/100 g. Different letters on columns represent statistically different mean values ($p \leq 0.01$).

The maximum dough height, $H_m$, for 0.5% *w/w* Rp is lower than the control, then increases with significant differences ($p \leq 0.01$) up to 2.0% *w/w* Rp and then decreased. According to the Tukey HSD test, the mean values of $H_m$ are statistically different ($p \leq 0.01$), except for the control and the sample with 1.0% Rp addition.

The time $T_1$, required to reach $H_m$, equals the duration of dough fermentation with Rp addition: 180 min for 1.5% *w/w*, 2.0% *w/w* and 2.5% *w/w* Rp and is slightly lower for the other samples with Rp and AA addition (Table 2). The time $T_1$ of the control is only 96 ± 1.41 min due to a fall of the dough with 5.73 ± 0.01% and is correlated with the time of relative stabilization related to the maximum point, $T_2$ and $T_2'$.

The dough height, $h$, when the fermentation ended shows the same evolution as $H_m$, being lower than $H_m$ only for the control, otherwise having values almost identical to $H_m$. The evolution of $H_m$ and $h$ of all the mixtures of WF with Rp and AA means that the obtained doughs are stable. The dough stability is demonstrated by the absence of the time of relative stabilization for all Rp and AA additions. Additionally, zero and low values of the percentage of fall, $(H_m - h)/H_m$, in %, sustain the dough stability.

Cao et al. [53] substituted the WF with 10–50% potato pulp from potatoes with 0.2% AA (*w/w*). They observed a bad influence on the dough development embodied in the significant reduction in the $H_m$ with substitutions higher than 30% potato pulp. For instance, the $H_m$ decreased by 58% from 28.0 mm (control) to 16.3 mm (50% potato pulp). Other authors [54] investigated the effect of exogenous salt, lactic acid, fat, and trehalose on the rheofermentographic parameters. They found that dough height was negatively influenced

by increasing salt and lactic acid additions and positively by moderate fat use. At the same time, trehalose alone could not affect the dough height.

### 3.3.2. Properties from the Gas Production Curve

The gas production curve is obtained through a direct and an indirect cycle. In the direct cycle, the rheofermentometer measures the gas production during the whole fermentation process, i.e., the porosity of the dough [44]. The properties of dough obtained from this curve are the height at the peak of the curve ($H_m'$, in mm), the time necessary to attain $H_m'$ ($T_1'$, in min), the moment when the gas starts to be lost from the dough ($T_x$, in min), and the volume of total gas produced during proofing (total volume, $V_t$, in mL). In the indirect cycle, the device measures the gas retention or porosity of the dough, i.e., the volume of $CO_2$ that escaped from the dough during proofing (lost volume, $V_l$, in mL). The difference between the two gas volumes represents the volume of gas ($CO_2$) kept in the dough (retained volume, $V_r$, in mL). Additionally, the rheofermentogram provides the coefficient of retention, *CR*, calculated with Equation (1):

$$CR = (V_r/V_t)\cdot 100, \text{ in } \% \tag{1}$$

The results obtained from the gas production curve are presented in Table 3 ($H_m'$, $T_1'$, $T_x$, and *CR*) and Figure 5 ($V_t$, $V_l$, and $V_r$).

**Table 3.** Rheofermentographic parameters from the gas formation and retention curve.

| Parameters<br><br><br>Samples | Maximum Height, $H_m'$, mm | Time to Reach $H_m'$, $T_1'$, min | Time of Porosity Appearance, $T_x$, min | Coefficient of Retention, *CR*, % |
|---|---|---|---|---|
| WF (Control) | 53.6 ± 0.14 [a] | 100 ± 1.41 [a] | 69.0 ± 1.41 [a] | 87,6 ± 0.08 [b] |
| WF–Rp 0.5% *w/w* | 53.9 ± 0.14 [a,b] | 114 ± 1.41 [b,c] | 71.0 ± 1.41 [a,b] | 91.5 ± 0.09 [f] |
| WF–Rp 1.0% *w/w* | 55.4 ± 0.14 [b,c] | 115 ± 2.10 [b,c] | 79.0 ± 0.70 [c] | 90.0 ± 0.21 [d] |
| WF–Rp 1.5% *w/w* | 56.7 ± 0.14 [c,d] | 118 ± 0.70 [c,d] | 83.0 ± 0.00 [c] | 91.0 ± 0.01 [e] |
| WF–Rp 2.0% *w/w* | 57.5 ± 0.70 [d] | 121 ± 0.70 [d,e] | 78.0 ± 1.41 [b,c] | 89.3 ± 0.08 [c] |
| WF–Rp 2.5% *w/w* | 56.6 ± 0.14 [c,d] | 123 ± 1.41 [d,e] | 76.5 ± 2.10 [a,b,c] | 91.0 ± 0.09 [e] |
| WF–AA 2 mg/100 g | 54.9 ± 0.14 [a,b] | 119 ± 0.70 [c,d] | 80.0 ± 1.41 [c] | 88.3 ± 0.07 [a] |

WF—wheat flour, Rp—rosehip powder, WF–Rp—mixtures of wheat flour with rosehip powder, WF–AA—wheat flour with ascorbic acid addition of 2 mg/100 g. Different letters on columns represent statistically different mean values ($p \leq 0.01$).

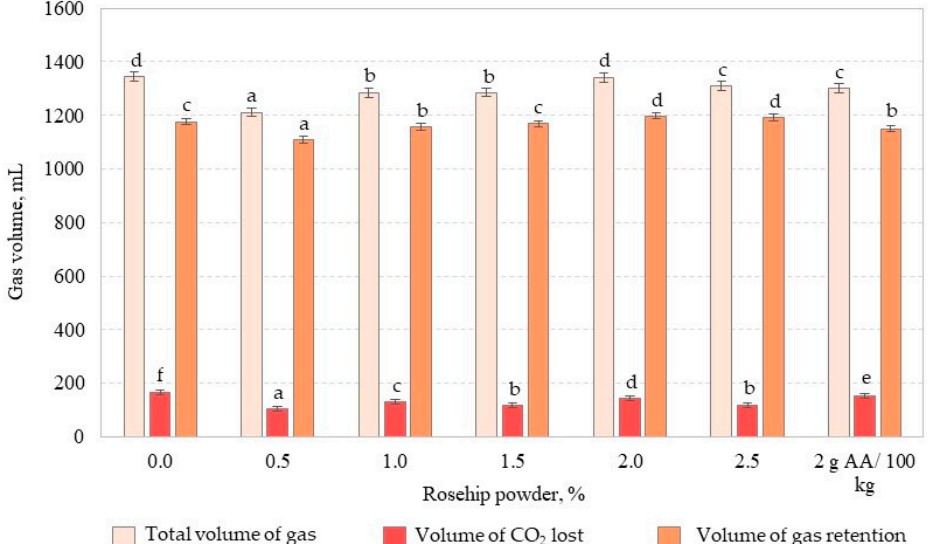

**Figure 5.** Gas volume as a function of Rp addition. Different lower-case letters represent statistically different mean values ($p \leq 0.01$). AA is ascorbic acid.

The maximum height $H_m'$ (mm) of the dough (Table 3) increases from $53.6 \pm 0.14$ mm (control) to $57.5 \pm 0.70$ mm (2.0% *w/w* Rp), then decreases. The time $T_1'$ increases slowly with Rp addition. The time $T_x$ of porosity appearance in the dough shows a similar evolution to $H_m'$, rising from $69.0 \pm 1.41$ min (control) to $83.0 \pm 0.00$ min (1.5% *w/w* Rp), then decreasing. All the parameters of the sample with 2 mg/100 g AA addition have values comparable to the samples with 1.0–2.0% *w/w* Rp, which indicates a similar behavior.

Moreover, the results mean that substituting WF with small percentages of 1.0–1.5% *w/w* Rp provides enough AA from this natural source to preserve the gluten network and allow obtaining raised bread with appropriate porosity.

The volume of gas given off during proofing increases from 1212 mL (0.5% *w/w* Rp) to 1342 mL (2.0% *w/w* Rp), then decreases. Additionally, the gas allowed to escape from the dough during proofing varies between 104 mL (0.5% *w/w* Rp) and 144 mL (2.0% *w/w* Rp). Thus, the gas retained within the dough during the whole process increases with the Rp addition. The control sample shows the highest total volume. However, it presents the highest volume of gas lost and the lower gas retention. This behavior is followed by the sample with AA addition (Figure 5). These results prove the contribution of Rp addition to improving the gluten network and increasing the volume of gas retention in the dough.

The *CR* of the samples with Rp and AA has higher values than the control (Table 3). *CR* is an accurate indication of the quality of gluten because a stronger gluten network retains more gas produced by fermentation [55]. Additionally, it reflects the gas retention capacity of the dough, which is essential in breadmaking because it affects the amount of gas produced and lost and the quality of bread [56].

Several other studies have investigated the influence of specific additions to WF on the production of gas during fermentation. Sun et al. [57] obtained an increase in gas production when WF was substituted with 5% and 10% maize gluten feed (MGF) and fermented maize gluten feed (FMGF) and decreased for higher levels of substitution, 15% and 20%, of the same ingredients. The authors of [53] reported a significant increase in the total gas volume and, at the same time, a decrease in the gas retention coefficient for adding 30% potato pulp from potatoes with 0.2% AA (*w/w*). Thus, potato pulp weakens the gluten network and negatively affects gas holding capacity. Liu et al. [58] added potato flour in proportions of 10%, 15%, 20%, 25%, 30% and 35%. They obtained an increase in the total $CO_2$ volume for 10% and 15% potato flour compared to the control, after which the values decreased for higher additions. The coefficient of gas retention was very high: 98.6% for the control and 98.1–98.7% for the samples with potato flour addition. Other authors [59] reported a decrease in the volume of gas retained in the dough due to adding bran with different particle sizes.

In conclusion, adding 0.5–2.5% *w/w* Rp in WF positively affects the rheofermentographic properties, increasing the height of the dough, the volume of gas retained in the dough, and the coefficient of retention.

### 3.4. Sensory Analysis of Bread

Figure 6 presents the photos of all the bread samples. The pictures are taken in cross-section through the bread. They give an image of the physical appearance of bread, e.g., the height, shape, and volume of the loaf of bread and the color of the crumb. Thus, it could be observed that bread with 1.5 *w/w* Rp has a higher height compared to all the other samples. Additionally, the color of the crumb intensifies with the increase in Rp values.

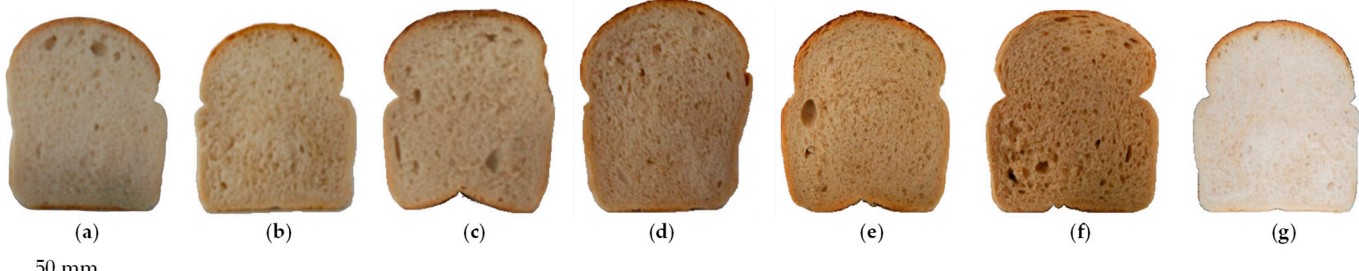

(a)          (b)          (c)          (d)          (e)          (f)          (g)

50 mm

**Figure 6.** Images of bread samples: (**a**) control; (**b**) bread with 0.5% *w/w* Rp; (**c**) bread with 1.0% *w/w* Rp; (**d**) bread with 1.5% *w/w* Rp; (**e**) bread with 2.0% *w/w* Rp; (**f**) bread with 2.5% *w/w* Rp; (**g**) bread with 2 mg AA/100 g.

The 20-point sensory test indicates a good acceptance of all the bread samples. Figure 7 presents the scores of sensory properties of evaluated bread The panel of assessors most appreciated the shape and volume, crumb appearance, flavor, and taste. Thus, the average scores of the shape and volume were 2.9 for bread with 0.5% *w/w*, 1.0% *w/w*, and 1.5% *w/w* Rp and 2.8 for the other bread from a maximum of 3 points. The bread with 1.5% *w/w* Rp obtained the highest scores for the flavor (3 points) and taste (3.8 points). Additionally, the crumb appearance received 3.6 points for bread with 1,5% *w/w*, 2.0% *w/w*, and 2.5% *w/w* Rp, 3.3 points for the control, and 3.5 points for the other bread. The other sensory attributes received good average scores, i.e., 2.5–2.7 for crust appearance and 2.6–3.0 for crumb consistency and chewing behavior. Although the Rp addition varies between 0.5% *w/w* and 2.5% *w/w*, the amounts being set primarily according to the AA requirement in the dough, the bread samples differ in crust appearance, crumb appearance, crumb structure and volume. However, the differences are not very large, as the results show. Therefore, the bread samples are close regarding sensory behavior (Figure 7).

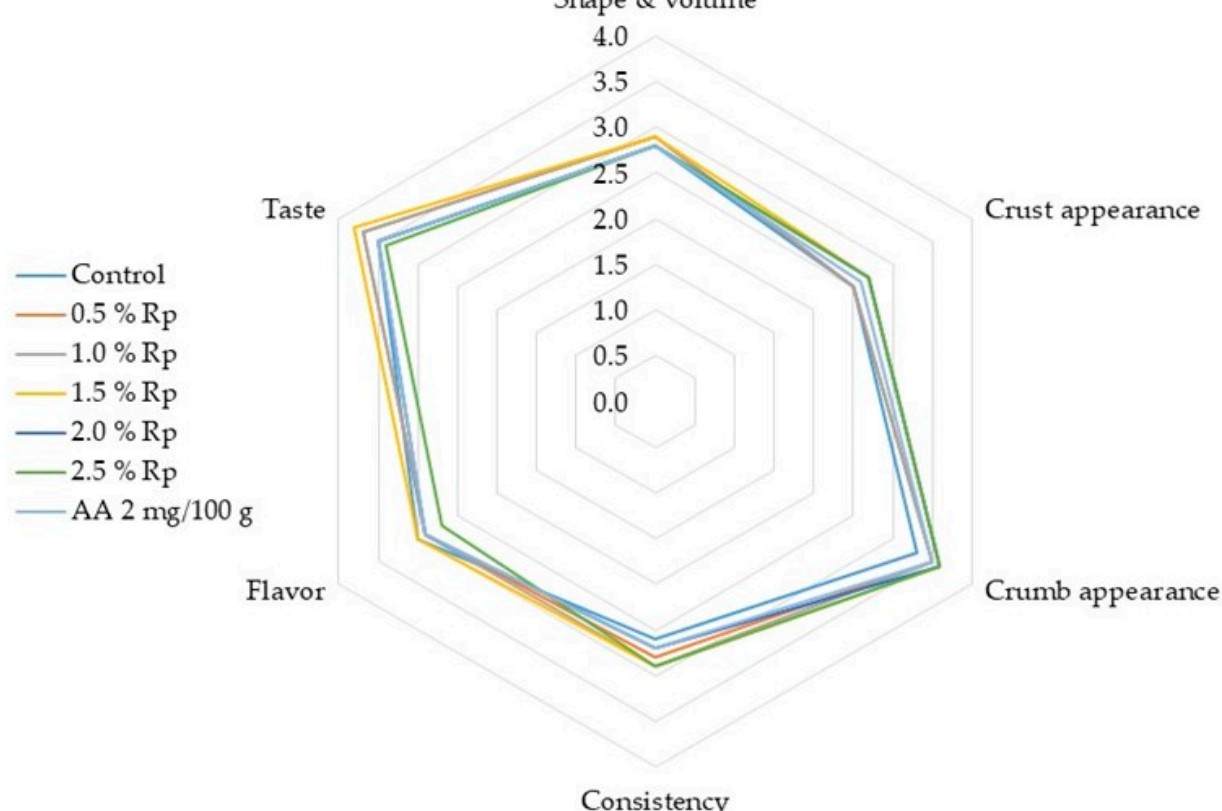

**Figure 7.** Sensory profile of bread with Rp addition based on the 20-point test.

Regarding the total score (Figure 8), all the bread samples obtained high values, the lowest one for control bread (17.7 points) representing 88.5% of 20 points. Thus, the bread with 1.5% $w/w$ Rp scored 18.9 points, followed by bread with 0.5% $w/w$ Rp (18.6 points), 1.0% $w/w$ Rp (18.4 points), 2.0% $w/w$ Rp (18.2 points) and 2.5% $w/w$ Rp (18.1 points). According to ANOVA, the differences are significant ($p < 0.05$). The bread with an addition of 2 mg AA/100 g scored 18.0 points and is situated between the control and samples with Rp addition.

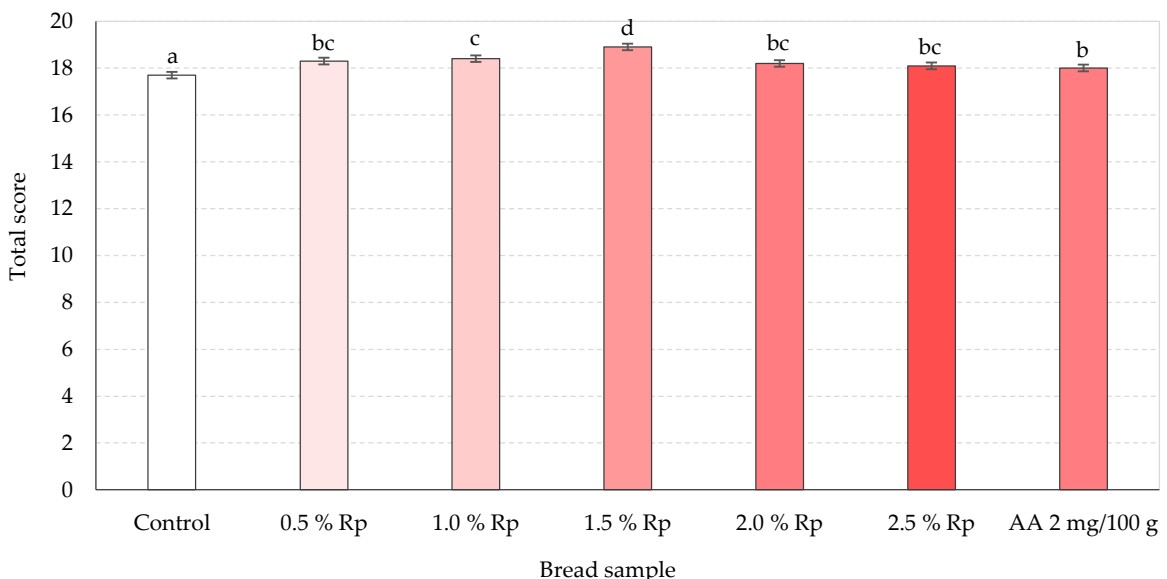

**Figure 8.** The total score of the 20-point sensory test. Different lower-case letters represent statistically different mean values ($p \leq 0.01$). Rp—rosehip powder, AA—ascorbic acid.

The higher total score of bread with Rp and AA additions compared to control bread reveals the acceptance and appreciation of bread by all the assessors. The findings are related to the presence of AA, either natural from Rp or synthetic, in the dough. As shown before, the AA improves the rheological properties of the dough and allows to obtain bread with better physicochemical properties. Additionally, the higher total score of bread samples with Rp indicates that the assessors consider them as having better taste, flavor, color and appearance. Thus, the color nuances of the crust and crumb with a reddish tint due to the carotenoid pigments Rp contains did not bother the assessors because they are used with whole meal or dark bread on the market.

Several other studies have applied sensory analysis to bread to investigate sensory properties and establish the degree of acceptance. Steffolani et al. [60] used a nine-point test to evaluate bread with an optimal addition of 10% chia seeds and flour. The sensory attributes scored were appearance, flavor, texture, aroma, aroma persistence, and overall acceptability. The evaluation of bread with pre-hydrated chia flour revealed insignificant differences compared to the control bread. In contrast, bread with non-hydrated chia flour received lower scores. However, the consumers preferred bread with chia seeds due to its similar appearance to bread with additions such as bran or different seeds and to the unusual greyish color of the crumb in bread with chia flour.

In other studies, when adding teff flour to WF up to 20% [61] and up to 30% [62,63], although the specific volume of the bread decreased and the firmness of the crumb increased, for additions of 5% and 10% sensory acceptability comparable to that of the control without additions was obtained. Even though the authors added enzymes to the higher proportion of teff addition, they found, applying a 10-point test, that the elasticity, softness, and brightness of the bread crumb decreased, and the bitter aroma and aftertaste increased when increasing the addition of teff [62]. Because the overall acceptability was significantly

lower ($p < 0.05$) for additions of 20% and 30% teff, only additions of 5 and 10% were considered acceptable.

Bernaert et al. [64] investigated the incorporation of 0.5%, 1.0%, 1.5%, and 2.0% air-dried and freeze-dried leek powder (*Allium ampeloprasum* var. *porrum*) in wheat bread. The sensory analyses showed that the attractivity of appearance, crumb color and odor decreased, the crust was darker, and the taste was less pleasant for higher concentrations of leek powder used in bread. Moreover, the results were more pronounced for adding freeze-dried leek powder than air-dried leek powder. The values of leek powder addition are similar to those of Rp used in the present work. However, the sensory attributes are very different, with bread with Rp addition being well accepted while bread with higher values of leek powder has lower acceptability.

In another recently published paper, Özcan [65] studied the effect of adding 0.5%, 1.0% and 1.5% ginger rhizome powder on the physicochemical, bioactive and sensory properties of bread. The author found that bread obtained by adding 0,5% ginger powder was most appreciated, while lower values of the flavor, color and crispness characterized bread with 1.5% ginger powder. However, the texture, porosity, and crispness of the bread with the 1.0% ginger powder addition were higher than for other bread samples.

It could be concluded that the sensory properties of bread with small additions of vegetal powders vary with the amount used, the powder properties, and the effects that powders can determine in bread. Anyway, the Rp addition to WF led to bread with good sensory properties and the acceptability of consumers. However, the bread with 1.0–2.0% $w/w$ Rp has the best sensory attributes.

## 4. Conclusions

The addition of Rp to WF aimed to replace synthetic ascorbic acid (AA) in breadmaking with a natural material rich in vitamin C. The Rp positively influenced the rheological properties of the dough, i.e., the extensographic, amylographic, and rheofermentographic properties. Thus, the dough resistance to extension, $R_{50}$, increased with Rp values; e.g., for the resting time of 90 min, the values were close to 1000 BU, the highest value recorded by the extensograph. The maximum resistance to extension ($R_{max}$) evolved similarly to the $R_{50}$ but with higher values. The extensibility decreased slowly with increased values of Rp, while the energy of the dough increased with 0.5%–1.5% $w/w$ Rp. A thorough analysis of the extensographic parameters showed that doughs with less than 2.0% $w/w$ Rp behaved better during the extensographic tests. The amylographic test showed that substituting WF with 0.5–2.5% $w/w$ Rp modified the gel properties and indicated good enzyme activity in the dough. The rheofermentographic examination revealed that the dough produced using flour mixtures with Rp addition was stable because the final height was close to the maximum. Dough stability was also sustained by the absence of the stabilization time relative to the maximum point of the dough development curve. The volume of gas retained in the dough increased by 0.5–2.0% $w/w$ Rp, leading to higher values of the retention coefficient. The panel of assessors highly appreciated the sensory properties of the bread, i.e., shape and volume, crust appearance, crumb appearance, flavor and taste. The sensory attributes of bread with Rp received higher total scores than the control bread.

Based on the results presented and discussed, the optimum concentration of Rp in this work was 1.5% $w/w$ for content in vitamin C of 6.0 $\pm$ 0.30 mg/100 g WF-Rp. However, the concentration of vitamin C in Rp at the moment of use could vary due to the source of Rp, the methods used to obtain Rp, and storage conditions. Additionally, the flour used could have different physicochemical properties. Therefore, an optimum range, e.g., 1.0–2.0% $w/w$ Rp, would be better. Moreover, it should be emphasized that the proximate composition of WF, Rp and WF-Rp mixtures, including the vitamin C content of Rp, should be periodically determined before use in practical applications (e.g., for batches, daily, etc.).

In conclusion, the study shows that the Rp could be a natural alternative to synthetic AA and its use in breadmaking could be appropriate.

## 5. Patents

The results were first used in a Romanian patent application no. "A/0069 of 19.09.2018": "Bread with rosehip powder addition and process for obtaining the same" ("Pâine din făină de grâu cu adaos de pudră de măceșe și procedeu de obținere a acesteia"). The abstract was published in "*Official Bulletin of Industrial Property, Patent Section*" ("*Buletinul Oficial de Proprietate Industrială, Secțiunea Brevete de Invenție*") no. 3/2020, p. 16.

**Author Contributions:** Conceptualization, N.V. and M.T.; Data curation, N.V. and M.T.; Formal analysis, N.V. and M.T.; Investigation, N.V.; Methodology, N.V. and M.T.; Project administration, M.T.; Supervision, M.T.; Validation, N.V. and M.T.; Visualization, N.V. and M.T.; Writing—original draft, N.V. and M.T.; Writing—review & editing, M.T. All authors have read and agreed to the published version of the manuscript.

**Funding:** Dunarea de Jos University of Galati funded the APC.

**Institutional Review Board Statement:** All the participants in the sensory panel signed an informed consent letter before participating in the study. The study was conducted according to the Declaration of Helsinki and the Ethics Committee of Dizing breadmaking company approved the protocol.

**Data Availability Statement:** The data presented in this study are available on request from the corresponding author. The data are not publicly available before the whole patent publication.

**Acknowledgments:** The authors thank Dizing SRL Brusturi, Neamt County, for technical support. This paper was supported by a grant offered by the Romanian Ministry of Research as an Intermediate Body for the Competitiveness Operational Program 2014-2020, call POC/78/1/2/, project number SMIS2014 + 136213, acronym METROFOOD-RO.

**Conflicts of Interest:** The authors declare no conflict of interest.

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
