# Peer review of "Effect of Rosehip Powder Addition on Dough Extensographic, Amylographic and Rheofermentographic Properties and Sensory Attributes of Bread"

_processes, doi:10.3390/pr11041088_

Round 1

Reviewer 1 Report

This manuscript is very interesting and it shows that rosehip powder in breadmaking may replace synthetic ascorbic acid based on dough extensographic, amylographic and rheofermentographic properties and sensory attributes of bread.

Author Response

Comment: This manuscript is very interesting and it shows that rosehip powder in breadmaking may replace synthetic ascorbic acid based on dough extensographic, amylographic and rheofermentographic properties and sensory attributes of bread.

Response: Thank you for the comments and appreciation of our manuscript.

Reviewer 2 Report

Reviewer comments

This manuscript presents an interesting study in the field of cereal science. The purpose of this study is to determine the effect of adding rosehip powder as a natural AA improver on the rheology and sensorial properties of enriched breads. In general, the work is good and original; however, the manuscript has some deficiencies due to language. Additionally, the reviewer has some questions, comments and suggestions to improve the manuscript:

Abstract

Why the authors use the word controls in the abstract, one with and one without AA, however, in the rest of the manuscript they considered just one control with 0% w/w Rp

Line 12:  better add w/w to unit of Rp addition: 0.5-2.5% w/w

Line 17: The authors compare the findings to a control: which one (with or without AA), please clarify this point also in the results section?

Which concentration of Rp powder is optimum in this work?

Introduction

References in introduction should be numbered as journal requirement.  Exp: Bourekoua et al. (2020) in line 58 and Franco et al. (2022) in line 64 and Vartolomei & Turtoi (2021) line 82.

Line 63: The authors cited Franco's reference without number at the beginning of the paragraph, and then cited the same reference numbered [12]; same comment in line 82:  reference number [26], please verify this point in the whole manuscript. There is no need to cite the same reference twice in one paragraph.

Line 85: correct 0.5–2.5% Rp addition, should be 0.5-2.5 % w/w because it is a replacement not an addition

The issue in the introduction is unclear. The authors must justify their choice of Rp as natural improver as well as the risks associated with synthetic AA.

 Materials and Methods

Line 95: Vartolomei & Turtoi (2021) should be numbred, please verify the whole manuscript

Line 99-104: the preparation of mixture must be added as a separate section: Formulation or preparation of mixture because it demonstrates a method rather than a material

Line 151: Breadmaking Procedure and Bread Characterization

Authors must describe the bread-making process and clarify the use of control with which formulation, even if it has previously been published. Because, the reader must have all of the necessary information in this manuscript, without having to return to the Vartolomei and Turtoi search each time.

Line 155: Why did the authors discuss the breads' physicochemical analysis if they did not present them in the results section?. Because, the goals of this work are only to evaluate rheology and sensorial attributes.

Line 136: in Rheofermentographic Measurements

The authors provided an analysis of the measurement with the parameters measured, but it lacks the method used, the step of measurement, and the dough preparation.

Results and Discussion

In Extensographic properties: the comparison to Gomez's work, which used different fiber materials, does not help the discussion at all

The same comment for the The ratio number discussion

The same remarks apply when comparing the gas production curve to Steffolani et al. (2015) with dough enriched with chia seeds.

The same comment for sensorial analysis: Line 531-548: We cannot compare the sensory characteristics of breads enriched with various plant materials because each plant material has a distinct taste, aroma, and color.

Line 403 : Gas production curve

The authors did not present the gas production curve, instead focusing on the numerical values. The word curve must be examined.

Figure 6: please add scale to images

Conclusions

The authors did not reach a final conclusion on the optimal concentration to use to improve both the rheology and sensory properties of Rp-fortified breads.

References:

As journal requirement, in the text, reference numbers should be placed in square brackets [ ].

Only, the reference list should be mentioned, as recommended by the ACS style guide

References list: authors must rewrite the references as journal requirement (ACS style)

Check reference 44 as journal requirement

Author Response

Answer to comments addressed by Reviewer 2

1. This manuscript presents an interesting study in the field of cereal science. The purpose of this study is to determine the effect of adding rosehip powder as a natural AA improver on the rheology and sensorial properties of enriched breads. In general, the work is good and original; however, the manuscript has some deficiencies due to language.

Response 1. Thank you for your positive statement and comments that helped us improve the manuscript.

Regarding the language, we would like to specify that the manuscript was written directly in English, so it was not translated from the native language of the authors. Then, it was double-checked with the Grammarly subscribed program and by an English professional. However, we agree that some language deficiencies may still exist in the manuscript. Therefore, after applying the corrections suggested/requested in the review process, we will check the English language in the manuscript again. Moreover, if the article is accepted for publication, there is still the proofreading stage in which the editorial team of the journal carries out fine-tuning of the English language.

As reviewers for other manuscripts, we know that reviewers are not supposed to correct the English language. However, when there is a statement such as „the manuscript has some deficiencies due to language,” a few examples would be helpful.

2. Additionally, the reviewer has some questions, comments, and suggestions to improve the manuscript:

Response 2. Thank you also for your questions, comments, and suggestions. All of them show how carefully you read the manuscript.

We were happy to answer all of them, ensuring the manuscript was improved this way.

Abstract

3. Why did the authors use the word controls in the abstract, one with and one without AA; however, in the rest of the manuscript, they considered just one control with 0% w/w Rp.

Response 3. We admit the mistake is ours and thank you for pointing it out.

The sentence „WF with and without AA addition of 2 mg/100 g were used as controls.” was replaced with: „WF without Rp or AA addition was used as control. A sample with an AA addition of 2 mg/100 g was also used.

4. Line 12: better add w/w to unit of Rp addition: 0.5-2.5% w/w

Response 4. The abbreviation „w/w” was added to the unit of Rp addition resulting in „0.5–2.5% w/w Rp” and specifically throughout the manuscript (e.g., 0.5% w/w Rp, 1.0% w/w Rp, etc.).

5. Line 17: The authors compare the findings to a control: which one (with or without AA)? Please clarify this point also in the results section.

Response 5. All the findings are compared to the control sample WF without Rp replacement or AA addition.

6. Which concentration of Rp powder is optimum in this work?

Response 6. We avoided to indicate an optimum concentration of Rp because the content of vitamin C in Rp at the moment of use could vary due to different factors. However, after a thorough analysis, we decided to do this. Therefore the optimum concentration of Rp was 1.5% w/w. However, to cover the variation in vitamin C content, we also indicate a range, such as 1.0%–2.0% w/w Rp.

Introduction

7. References in the introduction should be numbered as journal requirements.  Exp: Bourekoua et al. (2020) in line 58 and Franco et al. (2022) in line 64, and Vartolomei & Turtoi (2021) in line 82.

Line 63: The authors cited Franco’s reference without a number at the beginning of the paragraph and then cited the same reference numbered [12]; same comment in line 82:  reference number [26]. Please verify this point in the whole manuscript. There is no need to cite the same reference twice in one paragraph.

Response 7. The observation is fair. Indeed, there is no need to cite a reference twice in a paragraph. Moreover, the journal requirements indicate using numbers in square brackets, [ ], for citing the references. Corrections have been made throughout the manuscript.

However, the name of the first author has been kept when used at the beginning of a sentence, e.g., „Bourekoua et al. [11] studied” (according to https://www.mdpi.com/authors/english-editing: „If a reference is written at the beginning of a sentence, e.g. "[12] studied...", insert the author's name before the reference number, e.g. "Smith [12] studied" or write “The authors of [13] studied.”). This rule was not applied everywhere but only in several cases.

8. Line 85: correct 0.5–2.5% Rp addition. It should be 0.5-2.5 % w/w because it is a replacement, not an addition.

Response 8. The abbreviation „w/w” has been inserted: „0.5–2.5% w/w Rp”.

9. The issue in the introduction is unclear. The authors must justify their choice of Rp as a natural improver, as well as the risks associated with synthetic AA.

Response 9. The introduction has been restructured and the selection of Rp as a natural improver was better justified. Because there is no proper risk associated with synthetic AA, we used other reasons, such as the increase in demand for AA, consumer concerns about the use of food additives, etc.

The output of AA increased in the last decades and China became the world leader, providing around 80% of the quantity required for the food and beverages industry, pharmaceutical, beauty and personal care, and animal feed [17]. Synthetic AA received the status generally recognised as safe (GRAS) [7]. Also, the L-AA production and utilisation comply with good manufacturing practices (GMP) [7,18].

Nowadays, there is an increase in demand for AA, e.g., for the production of functional food and beverages [18]. Also, the concerns of consumers regarding the use of food additives and the consumption of food with high levels of synthetic additives determined the current trend to reduce the use or abandon artificial food additives and the consumer demand for natural [19], fresh, minimally processed foods [20], and foods with fewer additives and preservatives [21]. These trends led to the need to find other sources of AA, such as natural sources highly available [9]. Thus, research focused on replacing synthetic AA with AA-rich plant materials was initiated, disseminated, and applied in practice [22–24].

Materials and Methods

10. Line 95: Vartolomei & Turtoi (2021) should be numbered. Please verify the whole manuscript.

Response 10. „Vartolomei & Turtoi (2021)” has been replaced by [38]. The whole manuscript has been checked.

11. Line 99-104: the preparation of mixture must be added as a separate section: Formulation or preparation of mixture because it demonstrates a method rather than a material

Response 11. A new section, „2.2. WF-Rp mixtures preparation”, has been created.

12. Line 136: in Rheofermentographic Measurements

The authors provided an analysis of the measurement with the parameters measured, but it lacks the method used, the steps of measurement, and the dough preparation.

Response 12. The method used was mentioned in the first version of the manuscript with a proper citation: AACC method no. 89-01 [28] (L 139).

When a well-known method of analysis is used, its nomination in the research is sufficient. An in-depth description, including the steps of measurement, is necessary only if the method is slightly modified or adapted.

The manuscript is not the first presenting results obtained with a rheofermentometer. For instance, an article published in 2002 did not describe the method of analysis. The standard method is not mentioned, as one can see in the following quote:

The rheological properties of dough during fermentation were determined using a Rheofermentometer F2 (Tripette et Renaud, France), according to the supplier specifications.

The article is:

Wang, J.; Rosell, C.M.; de Barber, C.B. Effect of the addition of different fibres on wheat dough performance and bread quality. Food Chem. 79(2), 221–226. https://doi.org/10.1016/S0308-8146(02)00135-8

However, we inserted the information presented below to answer the reviewer.

The measurements included the following steps: yeast pretreatment, dough mixing, and gas production determination.

Dough formula. The formula of the dough was 100 WF of wheat mixtures, 2 g salt, 4.06 g compressed yeast, and warm distilled water. The water volume was correlated to the water absorption of flour: 58–62 mL/ 100 g flour [38]. The formula based on 100 g of flour can be upgraded to obtain a suitable quantity of dough.

Pretreatment of yeast. The compressed yeast was weighed accurately, crumbled into a beaker, and rehydrated in 25 mL of distilled water at 21 ± 1 °C five (5) min before mixing.

Mixing of dough. The dough was prepared in the bowl of a dough mixer (Tower T12039 Rose Gold, 2.5 L, 300 W). All dry ingredients were weighed, added to the bowl, and mixed with the yeast suspension. The remaining water was used to rinse the beaker and poured into the bowl. The water temperature was adjusted to obtain the dough at 30 ± 1 °C. The dough was mixed for 8–10 min until full development, then rested for 5 min while the temperature was measured.

Determination of gas production. The dough was introduced in an airtight container and subjected to fermentation in a temperature controlled environment (30 ± 1 °C). The test duration was 180 minutes. Gases were produced during fermentation due to yeast activity. The pressure inside the container was measured every 45 seconds. Also, a sensor above the dough showed its development and stability to determine the optimum baking time. Thus, a single automated test lasting for three hours (180 min) determined dough development, gas production, dough porosity, and dough tolerance during proofing. The results were delivered under a diagram called rheofermentogram, which contains two curves and all the calculations. The upper one, the dough development curve, reveals the maximum dough height (Hm), time to reach the maximum height (T1), final dough height (h), and time of relative stabilisation. The lower one, the gas release curve, was obtained through a direct cycle, in which the device measured the total gas production, and an indirect cycle, which measured the gas retention, i.e., the porosity of the dough [47].

We hope this does not lead to a higher similarity index.

13. Line 151: Breadmaking Procedure and Bread Characterization

Authors must describe the bread-making process and clarify the use of control with which formulation, even if it has previously been published. Because, the reader must have all of the necessary information in this manuscript, without having to return to the Vartolomei and Turtoi search each time.

Response 13. The answer is somewhat similar to response 12. Thus, if we repeat the breadmaking process description, the similarity index will increase no matter how much we would reword.

14. Line 155: Why did the authors discuss the breads' physicochemical analysis if they did not present them in the results section? Because the goals of this work are only to evaluate rheology and sensorial attributes.

Response 14. There is no proper discussion on the physicochemical properties of bread. It is only a referral to the previous article where the reader can find them.

Results and Discussion

15. In Extensographic properties: the comparison to Gomez's work, which used different fibre materials, does not help the discussion at all

16. The same comment for the The ratio numberdiscussion

Responses 15+16. We selected works using vegetal ingredients to substitute wheat flour for the discussion.

Gómez et al. [45] used fibres from peas, cocoa, coffee, oranges, wheat, and microcrystalline cellulose. The values used are close (2% and 5%) and the results are similar. 

Gómez et al. [45] was the only proper work to be considered for discussion on R50 and Rmax.

Thus, we consider the discussion useful.

17. The same remarks apply when comparing the gas production curve to Steffolani et al. (2015) with dough enriched with chia seeds.

Response 17. We eliminated Steffolani et al. and introduced Sun et al. [57]. However, Steffolani enriched the dough with chia seeds and chia flour.

18. The same comment for sensorial analysis: Line 531-548: We cannot compare the sensory characteristics of breads enriched with various plant materials because each plant material has a distinct taste, aroma, and color.

Response 18. We agree that every addition of plant materials gives the bread specific/distinct sensory attributes (taste, aroma, colour, etc.). However, we consider the discussion right since we did not analyse the compounds giving distinctive flavours or aromas. Instead, we only discussed the perceptions of the assessors and the levels of appreciation of bread.

19. Line 403: Gas production curve

The authors did not present the gas production curve, instead focusing on the numerical values. The word curve must be examined.

Response 19. This observation is fair and subtle. Thank you.

All the rheological tests – extensographic, amylographic and rheofermentographic – have as result diagrams (named extensogram, amylogram and rheofermentogram). These diagrams consist of different curves because the measurements are done continuously at certain small intervals (e.g., 45 s). The soft of each device calculates and gives the values of specific parameters.

Concerning the rheofermentographic measurements, the standard method, the specifications of the device supplier, and the literature present two major curves: the dough development curve and the gas production curve.

To comply with the requirements of the reviewer, we reworded and discussed the properties resulting from each curve:

3.3.1. Properties from the dough development curve

3.3.2. Properties from the gas production curve

This approach does not change the discussion much, but it does clarify it.

20. Figure 6: please add scale to images

Response 20. The scale was added to the photos.

Conclusions

21. The authors did not reach a final conclusion on the optimal concentration to use to improve both the rheology and sensory properties of Rp-fortified breads.

Response 21. The observation is fair, thank you.

We added partial conclusions to each test (extensographic, amylographic, rheofermentographic), sensory analysis, and a principal conclusion at the end.

References

22. As journal requirement, in the text, reference numbers should be placed in square brackets [ ].

Only the reference list should be mentioned, as recommended by the ACS style guide

Response 22. Please see the Answer 7.

23. References list: authors must rewrite the references as journal requirement (ACS style)

Response 23. The references were checked and corrected.

24. Check reference 44 as journal requirement

Response 24. The reference was corrected.

The entire manuscript was revised following the suggestions and recommendations of all reviewers.

Thank you

Reviewer 3 Report

This article is devoted to the Effect of rosehip powder addition on dough extensographic, 2 amylographic and rheofermentographic properties and sensory 3 attributes of bread. The study showed that the Rosehip used in breadmaking to replace synthetic Ascorbic acid could be appropriate.The article is written at a high level, the amount of data and the subject correspond to the subject and requirements of the journal. The results were first used in a Romanian patent application no. A/0069 of 19.09.2018: 579 Bread with rosehip powder addition and process for obtaining the same.

The authors publised titled The Influence of the Addition of Rosehip Powder to Wheat Flour on the Dough Farinographic Properties and Bread Physico-Chemical Characteristics at Applied Sciences. This paper is also complementary of this published paper.

There are some points that could be improved:

1. Introduction can be improved.

2. Please add more literature comparisons. This will expand the description of the results and lead to greater validity of the conclusions.

3. Please add letters at the Figures for Sensory analysis discussion section 

Author Response

Author's Reply to the Review Report (Reviewer 3)

This article is devoted to the Effect of rosehip powder addition on dough extensographic, amylographic and rheofermentographic properties and sensory attributes of bread. The study showed that the Rosehip used in breadmaking to replace synthetic Ascorbic acid could be appropriate. The article is written at a high level, the amount of data and the subject correspond to the subject and requirements of the journal. The results were first used in a Romanian patent application no. A/0069 of 19.09.2018: 579 Bread with rosehip powder addition and process for obtaining the same.

The authors published titled The Influence of the Addition of Rosehip Powder to Wheat Flour on the Dough Farinographic Properties and Bread Physico-Chemical Characteristics at Applied Sciences. This paper is also complementary of this published paper.

Thank you for your positive statement and comments that helped us improve the manuscript.

There are some points that could be improved:

1. Introduction can be improved.

Response 1. The introduction has been improved following the suggestions and recommendations of reviewers. Rewordings were applied where the Ithenticate Report indicated so. Also, the section has been restructured and the selection of Rp as a natural improver was better justified, as another reviewer suggested.

2. Please add more literature comparisons. This will expand the description of the results and lead to greater validity of the conclusions.

Response 2. New references were added and used in the Introduction and Results and Discussion sections:

  1. Smirnoff, N. Ascorbic acid metabolism and functions: A comparison of plants and mammals. Free Radic. Biol. Med. 2018, 122, 116–129. [https://doi.org/10.1016/j.freeradbiomed.2018.03.033]
  2. Hon, S.L. Vitamin C. In Encyclopedia of Toxicology, vol. 4, 3rd ed.; Wexler Ph., Editor-in-Chief; Elsevier, Academic Press: London, UK, 2014; pp. 962–963.
  3. Chang, S.K.; Ismail, A; Daud Z.A.M. Ascorbic Acid: Properties, Determination and Uses. In Encyclopedia of Food and Health. Caballero B., Finglas P.M., Toldrá F., Eds.; Elsevier, Academic Press: Oxford, 2016; pp. 275 – 284.
  4. Commission Regulation (EU) No 1129/2011 of 11 November 2011 amending Annex II to Regulation (EC) No 1333/2008 of the European Parliament and of the Council by establishing a Union list of food additives. Official Journal of the European Union. 2011, L295, 1–177. Available online: https://eur-lex.europa.eu/legal-content/EN/TXT/PDF/?uri=CELEX:32011R1129&from=EN (accessed on 17 March 2023).
  5. US Food and Drug Administration. ”Food Additive Status List.” 2012, Available online: https://web.archive.org/web/20120117060614/https://www.fda.gov/Food/FoodIngredientsPackaging/FoodAdditives/FoodAdditiveListings/ucm091048.htm (accessed on 17 March 2023).
  6. Health Canada. ”List of Permitted Preservatives (Lists of Permitted Food Additives).” Government of Canada. 2006, Available online: https://www.canada.ca/en/health-canada/services/food-nutrition/food-safety/food-additives/lists-permitted/11-preservatives.html (accessed on 17 March 2023).
  7. Australia New Zealand Food Standards Code – Revocation and Transitional Variation 2015 (Application A1101 – Commencement of Dietary Fibre Claim Provisions – Consequential). Commonwealth of Australia Gazette. 2015, FSC 99. Available online: https://www.legislation.gov.au/Details/F2015L01391 (accessed on 17 March 2023).
  8. Vidit, G.; Roshan, D. Ascorbic Acid Market. Global Opportunity Analysis and Industry Forecast, 2021-2031. Report Code: A07444. Allied Market Research, 2022, 370 pages. Available online: https://www.alliedmarketresearch.com/ascorbic-acid-market-A07444, (accessed on 18 March 2023).
  9. Ascorbic acid, L- (300). Food additive details. GSFA Online. Codex alimentarius, FAO/WHO Food Standards. 2021. Available online: https://www.fao.org/gsfaonline/additives/details.html?d-3988876-o=2&id=241, (accessed on 18 March 2023).
  10. Mesías, F.J.; Martín, A.; Hernández, A. Consumers’ growing appetite for natural foods: Perceptions towards the use of natural preservatives in fresh fruit. Food Res. Int. 2021, 150(Part A), 110749. [https://doi.org/10.1016/j.foodres.2021.110749]
  11. Allende, A.; Tomás-Barberán, F.A., Gil, M.I. Minimal processing for healthy traditional foods. Trends Food Sci. Technol. 2006. 17(9), 513–519. [https://doi.org/10.1016/j.tifs.2006.04.005]
  12. Lavilla, M.; Gayán, E. Consumer acceptance and marketing of foods processed through emerging technologies. In Innovative Technologies for Food Preservation. Barba F.J., Sant’Ana A.S., Orlien V. & Koubaa M., Eds. Elsevier, Academic Press: London, UK, 2018; pp. 233–253. [https://doi.org/10.1016/B978-0-12-811031-7.00007-8]
  13. Gomathi, G.K.; Parameshwari, S. Evaluation of buckwheat flour addition on the sensory, nutritional and materialistic properties analysis of Indian flat bread. Mater. Today 2022, 66(Part 3), 988–995. [https://doi.org/10.1016/j.matpr.2022.04.778]
  14. Nikolić, N.Č.; Stojanović Krasić, M.; Šimurina, O.; Cakić, S.; Mitrović, J.; Pešić, M.; Karabegović, I. Regression analysis in examination the rheology properties of dough from wheat and Boletus edulis flour. J. Food Compost Anal. 2022. 115, 105022. [https://doi.org/10.1016/j.jfca.2022.105022]
  15. Hackenberg, S.; Verheyen, C.; Jekle, M.; Becker, T. Effect of mechanically modified wheat flour on dough fermentation properties and bread quality. Eur. Food Res. Technol. 2017, 243, 287–296. [https://doi.org/10.1007/s00217-016-2743-8]
  16. Sun, X.; Ma, L.; Zhong, X.; Liang, J. Potential of raw and fermented maize gluten feed in bread making: Assess of dough rheological properties and bread quality. LWT - Food Sci. Technol. 2022, 162, 113482. [https://doi.org/10.1016/j.lwt.2022.113482]
  17. Mohammed, M.I.O.; Mustafa, A.I.; Osman G.A.M. Evaluation of wheat breads supplements with Teff (Eragrostis tef ZUCC. Trotter) grain flour. Aust. J. Crop Sci. 2009, 3, 207–212.

3. Please add letters at the Figures for Sensory analysis discussion section

Response 3. Letters were added in Figure 8.

The entire manuscript was revised following the suggestions and recommendations of all reviewers.

Thank you

Round 2

Reviewer 2 Report

This manuscript presents an interesting study in the field of cereal science. The purpose of this study is to determine the effect of adding rosehip powder as a natural AA improver on the rheology and sensorial properties of enriched breads. In general, the work is good and original.

the authors have taken into consideration all the remarks of reveiwer and the manuscript is improved